# The Effects of Blood Flow Restriction Aerobic Exercise on Body Composition, Muscle Strength, Blood Biomarkers, and Cardiovascular Function: A Narrative Review

**DOI:** 10.3390/ijms25179274

**Published:** 2024-08-27

**Authors:** Chaeeun Cho, Sewon Lee

**Affiliations:** 1Department of Human Movement Science, Graduate School, Incheon National University, Incheon 22012, Republic of Korea; channy27@inu.ac.kr; 2Division of Sport Science, College of Arts & Physical Education, Incheon National University, Incheon 22012, Republic of Korea; 3Sport Science Institute, College of Arts & Physical Education, Incheon National University, Incheon 22012, Republic of Korea; 4Health Promotion Center, College of Arts & Physical Education, Incheon National University, Incheon 22012, Republic of Korea; 5Research Center of Brain-Machine Interface, Incheon National University, Incheon 22012, Republic of Korea

**Keywords:** cardiovascular function, elderly, fat mass, glycemic metabolism, lipid profile, muscle mass, obesity, sarcopenia

## Abstract

Blood flow restriction exercise has emerged as a promising alternative, particularly for elderly individuals and those unable to participate in high-intensity exercise. However, existing research has predominantly focused on blood flow restriction resistance exercise. There remains a notable gap in understanding the comprehensive effects of blood flow restriction aerobic exercise (BFRAE) on body composition, lipid profiles, glycemic metabolism, and cardiovascular function. This review aims to explore the physiological effects induced by chronic BFRAE. Chronic BFRAE has been shown to decrease fat mass, increase muscle mass, and enhance muscular strength, potentially benefiting lipid profiles, glycemic metabolism, and overall function. Thus, the BFRAE offers additional benefits beyond traditional aerobic exercise effects. Notably, the BFRAE approach may be particularly suitable for individuals with low fitness levels, those prone to injury, the elderly, obese individuals, and those with metabolic disorders.

## 1. Introduction

The American College of Sports Medicine (ACSM) recommends moderate to high-intensity aerobic exercise (46–90% VO_2_ max) for 3–7 days per week to enhance cardiopulmonary function, improve metabolism, and reduce the risk of cardiovascular disease (CVD) [1]. High-intensity aerobic exercise, such as high-intensity interval training (HIIT), is known for its capacity to enhance exercise capacity and various health outcomes [2]. However, it may pose the risk of high mechanical stress and injury, limiting its applicability to certain populations, including older adults. To address this concern, blood flow restriction exercise (BFRE), also known as “KAATSU” exercise, has emerged as an effective intervention for individuals who may find high-intensity exercise challenging. BFRE involves applying pressure to the limbs during exercise using air cuffs, which restricts blood flow to the exercising muscles and induces a hypoxic environment within the muscle tissue [3]. BFRE can be integrated with various exercise modalities, including resistance training, walking, jogging, and cycling. BFRE can be categorized into two primary types: blood flow-restricted resistance exercise (BFRRE) and blood flow-restricted aerobic exercise (BFRAE). In both BFRRE and BFRAE, a cuff is applied to either the upper thigh or upper arm, depending on whether the exercise targets the upper or lower extremities. Specifically, for upper extremity exercises, the cuff is placed on the upper arm, while for lower extremity exercises, it is positioned on the upper thigh. In the context of BFRAE, most exercises are performed with the cuff placed on the upper thigh.

BFRE is commonly combined with low-intensity exercise and has demonstrated effectiveness in increasing muscle mass and strength with relatively light loads [4]. Traditionally, muscle strength and hypertrophy are achieved through high-intensity resistance exercise at approximately 70–85% of 1 Repetition Maximum (1RM). However, previous studies have shown that six weeks of BFRRE at 20% of 1RM resulted in strength gains comparable to those achieved by a load of 80% of 1RM without blood flow restriction [5,6]. Recent studies have reported that acute BFRRE can promote muscle hypertrophy by inducing increases in lactate, growth hormone (GH), insulin-like growth factor 1 (IGF-1), and testosterone immediately post-exercise, compared to resistance exercise without blood flow restriction [4,7]. This suggests that combining low-intensity exercise with blood flow restriction could activate mechanisms conducive to muscle hypertrophy, which may be particularly beneficial for individuals who face challenges with high-intensity aerobic exercise. As a result, BFRE combined with low-intensity exercise has been recognized as an effective form of exercise in geriatrics, athletic training, and rehabilitation [2]. 

Building on the positive outcomes observed with BFRRE, recent research has explored BFRAE. Eight weeks of BFRAE at low intensity (40% VO_2_ max) resulted in increased strength and muscle mass compared to a group without blood flow restriction, and six weeks of low-intensity BFRAE (30% heart rate reserve [HRR]) increased leg lean mass, although no significant difference was observed in either the control group or the high-intensity (60–70% HRR) exercise group without blood flow restriction [8,9]. These findings suggest that BFRAE can yield benefits such as increased muscle mass and strength, even in the absence of traditional resistance exercise. BFRAE is often implemented at low intensity and may serve as an effective exercise strategy for populations with low fitness levels or heightened injury risks. Furthermore, BFRAE has the potential to provide additional physiological benefits related to cardiovascular function, lipid and glucose metabolism, and body composition, although these advantages have not been extensively documented. 

To date, while these findings suggest the favorable effects of BFRAE, the majority of blood flow restriction studies have focused on resistance exercise [10,11]. However, resistance exercise aimed at muscle hypertrophy and strength gains entails substantial biomechanical stress and an elevated risk of musculoskeletal injury. Therefore, it is pertinent to explore the potential of BFRAE and its effectiveness as an intervention strategy across various populations. Regrettably, the quantity of studies that have investigated the physiological advantages of long-term BFRAE to date has been insufficient for a systematic review. As a result, this review is presented in a narrative format to integrate the studies that have reported physiological benefits of BFRAE to date. The objective of this review is to provide a comprehensive overview that enhances the current understanding of BFRAE’s effects. Specifically, this review aims to summarize recent evidence regarding the impact of BFRAE on body composition, muscle strength, and glucose metabolism, and cardiovascular function.

## 2. Methods

A comprehensive literature search was conducted using the databases PubMed, Web of Science, and Google Scholar. The search keywords were employed in various combinations including “blood flow restriction aerobic exercise or BFRAE” AND “treadmill or cycle or walking or running or aerobic exercise” AND “body fat or fat mass or body fat percent or body weight or visceral fat or circumference” AND “muscle mass or muscle cross sectional area or muscle strength or muscle volume or muscle hypertrophy or lean mass” AND “glucose or insulin or cholesterol or triglyceride” AND “pulse wave velocity or blood pressure or vascular function or vascular compliance or vascular stiffness or flow mediated dilation.” The inclusion criteria for this review focused on studies that investigated the effects of chronic BFRAE on body composition, muscle strength, glucose metabolism, and cardiovascular function, with a particular emphasis on clinical trials conducted in human subjects. The exclusion criteria encompassed the omission of animal studies, longitudinal studies, and meta-analyses as these were beyond the scope of this review. Additionally, studies that did not precisely describe the exercise intervention or studies that only treated simple blood flow restriction without exercise were excluded. 

The Scale for the Assessment of Narrative Review Articles (SANRA) was utilized to ensure the quality of the narrative review [12]. The scale consists of six items that range as follows: 0 (low quality), 1 (intermediate quality), and 2 (high quality). It encompasses the following areas: descriptions of (1) the significance and (2) the objectives of the article, (3) research in the existing literature and (4) citation methods, and presentation of (5) the evidence quality and (6) pertinent endpoint information. The total score for this review was calculated as 12 points.

## 3. Potential Physiological and Molecular Mechanisms of Blood Flow Restriction Exercise

Blood flow restriction has primarily been associated with resistance exercise and has been shown to induce muscle hypertrophy and increases in strength, even at low intensities [13,14]. However, the precise mechanisms underlying these effects remain unclear. Blood flow restriction during exercise can stimulate a variety of physiological responses and molecular signaling pathways, which are thought to be due to venous pooling caused by the partial restriction of blood flow. This can lead to potential adaptive responses in the muscle, including hormone secretion, accumulation of metabolites, and cell swelling due to blood flow restriction [15].

Blood flow restriction during exercise can elicit a state of metabolic stress, characterized by the accumulation of metabolites, such as lactate and inorganic phosphate, a decrease in pH, and hypoxic conditions. This typically occurs during high-intensity exercise or in a hypoxic environment and is considered a crucial factor in promoting muscle hypertrophy [16]. These factors play a crucial role in muscle growth and are likely induced by blood flow restriction during exercise.

Previous studies have shown that increased concentrations of anabolic hormones, such as GH and IGF-1, following resistance exercise, are important mechanisms that can induce muscle hypertrophy [17,18]. The accumulation of metabolites during BFRE results in a decrease in pH, which can stimulate the secretion of GH [19]. Recent studies have reported significant increases in lactate, GH, and IGF-1 levels immediately after a single bout of low-intensity BFRRE compared to moderate-intensity resistance exercise [7,20]. Additionally, two weeks of low-intensity BFRRE resulted in increased IGF-1 concentrations, and a 7.7% increase in lower extremity muscle volume, compared to a 1.9% increase in a non-restricted resistance exercise group [21]. These findings demonstrate that blood flow restriction can stimulate an increase in GH and IGF-1 along with metabolite accumulation. 

Muscle growth can stimulate protein synthesis by activating signaling pathways related to muscle synthesis [22,23]. Among these signaling pathways, mammalian target of rapamycin (mTOR) is considered a key factor in regulating skeletal muscle growth. mTOR is involved in the regulation of messenger ribonucleic acid (mRNA) translation initiation and plays an important role in exercise-induced muscle protein synthesis and training-induced hypertrophy [24]. Recent studies have reported increases in ribosomal S6 kinase 1 (S6K1) phosphorylation and ribosomal protein S6 (rpS6) phosphorylation, which are downstream targets of mTOR, during acute low-intensity (20% 1RM) BFRRE compared to a non-restricted resistance exercise group [25,26]. In addition, the increased concentration of Ca2+ during exercise can lead to the production of nitric oxide synthase 1 (NOS-1) and nitric oxide (NO), which can directly activate the mTOR signaling pathway, promoting protein synthesis [27]. In a study of the animal model, protein synthesis-related signaling pathways were investigated 24 h after the last session of 2 weeks of blood flow restriction. Compared to a non-restricted group, blood flow restriction treatment not only enhanced the signaling pathways that promote protein synthesis but also reduced the expression of the ubiquitin-proteasome and myostatin, which inhibit muscle degradation, thereby preventing muscle atrophy and promoting muscle hypertrophy [28].

Another possible mechanism by which BFRE may cause muscle hypertrophy is cell swelling. Temporary blood flow restriction causes an accumulation of metabolites, increasing the osmotic pressure difference between the inside and outside of the cell. This may lead to an influx of water into the cell, leading to cell swelling. Haussinger et al. reported that cell swelling can inhibit catabolism and stimulate anabolism [29]. Muscle swelling was significantly greater during BFRRE than during unrestricted resistance exercise, and acute BFRRE resulted in a post-exercise increase of 5.6% in transient femoral muscle volume, compared to about 4% increase in the high-intensity resistance exercise group [30,31]. Additionally, a study showed that low-intensity BFRRE induces changes in muscle swelling and plasma volume similar to those observed during high-intensity resistance exercise [32]. Thus, the temporary accumulation of blood resulting from blood flow restriction may lead to cellular edema, potentially stimulating muscle synthesis. However, limited research has been conducted on this mechanism, and there is insufficient evidence to determine whether the inhibition of catabolic and anabolic activity due to cell edema following BFRRE is advantageous for muscle hypertrophy.

Recent studies have confirmed that muscle hypertrophy and strength can increase in response to BFRAE. For instance, acute BFRAE performed at 40% of VO_2_ max resulted in a significant increase in serum GH levels in a blood-restricted walking group compared to a non-restricted exercise group, and IGF-1 serum concentrations were increased only in the blood-restricted walking group [32]. Furthermore, a study involving healthy young men reported an increase in serum GH immediately after blood-restricted aerobic walking compared to a group without blood restriction [33]. These findings suggest that BFRAE increases the circulating concentrations of anabolic hormones that induce muscle growth, potentially contributing to muscle hypertrophy and strength gains. 

In this respect, both BFRRE and BFRAE may induce muscle hypertrophy through an increase in anabolic hormones, suggesting that they may share partially similar mechanism. However, the signaling pathways related to muscle synthesis and the mechanisms of cell swelling induced by BFRAE are not yet fully understood, particularly in comparison to BFRRE. Given that resistance exercise and aerobic exercise with blood flow restriction are fundamentally different forms of exercise, they may involve distinct molecular mechanisms. Nonetheless, due to the commonality of blood flow restriction, these two types of exercise may share common mechanisms. However, the current evidence is still insufficient to make a clear conclusion regarding these similarities.

Taken together, muscle synthesis can be induced by a variety of complex mechanisms, including metabolic stressors such as metabolite accumulation, oxygen deprivation, cell swelling, and hormonal responses. Nevertheless, the majority of studies on mechanisms have focused on BFRRE, while this review’s emphasis is on BFRAE. In this regard, there is still a lack of evidence regarding BFRAE, and further physiological and molecular studies are needed to establish specific and clear mechanisms.

## 4. Effects of Blood Flow Restriction Aerobic Exercise on Body Composition

Obesity is closely associated with an increased risk of cardiovascular, metabolic, and cancer diseases and is a major cause of mortality worldwide [33,34,35]. Therefore, exploring effective exercise intervention strategies to reduce body fat mass is essential. However, obese individuals often face limitations with various exercise interventions due to their low physical fitness levels and medical history [36]. Recent studies have reported fat and body circumference reduction with BFRAE, and a comprehensive summary of these results is presented in Table 1.

The specific mechanism by which BFRAE can reduce body fat remains unclear, but several hypotheses can be proposed. Restricted blood flow to the exercising muscles leads to the accumulation of metabolic by-products in the blood, such as lactic acid and inorganic phosphate causing metabolic stress [37,38,39]. This accumulation of metabolic stress increases blood flow to the intracellular space of muscle cells, inducing cell swelling, which has been reported to enhance glycerol turnover, reflecting whole-body fat mobilization and increasing lipid breakdown [40,41,42]. However, additional research is necessary to demonstrate this, as direct fat metabolism markers were not measured in the context of BFRAE.

Previous studies have reported significant increases in energy expenditure and excess post-exercise oxygen consumption (*p* < 0.001 and *p* < 0.001, respectively) during BFRAE compared to unrestricted aerobic exercise, and Killian Salzmann et al. concluded that the combination of BFRAE and increased oxygen consumption in working muscles may heighten oxygen dependence [43,44]. Blood flow restriction leads to an increase in the venous fluid/pyruvate ratio, which in turn causes a shift to anaerobic metabolism and elevated ratings of perceived exertion, resulting in higher blood lactate levels and increased metabolic stress. This metabolic stress can potentially increase energy expenditure [45,46]. Increased physiological load may stimulate the production of fat-reducing hormones such as GH and catecholamine, potentially affecting the reduction in body fat [47]. Shuoqi Li et al. assessed body composition and GH concentrations at least 48 h after twelve weeks of HIIT with blood flow restriction (85% VO_2_ max) in obese subjects [48]. The study found a significant reduction in body fat of approximately 3% in the HIIT group with blood flow restriction compared to a 2% reduction in the HIIT group without blood flow restriction (85% VO_2_ max). Additionally, there was a concomitant significant increase in GH in the blood flow restriction group (*p* < 0.05), whereas no increase in GH was observed in the group without blood flow restriction. Furthermore, a study investigating the acute hormonal response to blood flow-restricted walking in older women reported an increase in adrenaline concentration (*p* = 0.016), and in healthy young men, a study reported an increase in GH (*p* < 0.01) immediately after a single bout of blood flow-restricted aerobic walking compared to a group without blood flow restriction [49,50].

Although evidence is currently limited, it is believed that forms of aerobic exercise that incorporate blood flow restriction may increase the production of sympathomimetic hormones, such as GH and catecholamines, which promote the breakdown of body fat. Consequently, BFRAE may contribute to body fat reduction by activating secretion of these hormones. However, there is a scarcity of research demonstrating whether changes in body composition resulting from chronic BFRAE are primarily due to fat oxidation during exercise or alterations in basal fat oxidation capacity. 

Taken together, BFRAE may be an effective exercise intervention for reducing body fat compared to traditional aerobic exercise, particularly in obese and overweight individuals. Nevertheless, the specific mechanisms underlying fat loss associated with BFRAE require further investigation and validation in non-obese and non-overweight populations. Therefore, while BFRAE appears to be potentially more effective than classical aerobic exercise for body fat reduction in obese and overweight individuals, further research is needed to confirm these findings and elucidate the underlying mechanisms. 

**Table 1 ijms-25-09274-t001:** Effects of aerobic exercise with blood flow restriction on body composition.

Author	Subjects	Group	Intervention	Cuff Pressure	Outcomes	*p*-Value
**Shuoqi Li et al. [48]**	Obese adults (n = 72, <25 yr, BF% > 30%)	CON HIIT HIIT + BFR (during interval) HIIT + BFR (during exercise)	Frequency: 12 wks, 2 times/wk Intensity: 85% VO_2_ max Volume: 4 sets (each set 3 min, 3 min rest) Type: HIIT	40% limb occlusive pressure (LOP)	%BF ↓ FM ↓ AVFA ↓	The HIIT + BFR (during interval) and HIIT + BFR (during exercise) groups showed a significant decrease in %BF compared to the HIIT group (*p* < 0.05). The HIIT + BFR (during interval) group showed a significant decrease in FM and AVFA compared to the HIIT group (*p* < 0.05).
**Yong Chen et al. [51]**	Obese men (n = 40, 18–22 yr, %BF > 25% or BMI > 28 kg/m^2^)	LIT LIT + BFR	Frequency: 12 wks, 2 times/wk Intensity: 40% VO_2_ max Volume: 3 sets (each set 15 min, 1 min rest) Type: Cycle	200 mmHg	BW ↓ WC ↓ FM ↓ %BF ↓	The LIT + BFR group showed significant decreases in BW, WC, FM, and %BF compared to the LIT group (*p* < 0.05).
**Amir Kargaran et al. [52]**	Older women (n = 24, 62.9 ± 3.1 yr)	CON DT DT + BFR	Frequency: 8 wks, 3 times/wk Intensity: 45% HRR Volume: 20 min Type: Walk	150–200 mmHg	BW ↓ Visceral fat ↓	The DT + BFR group showed significant decreases in BW (*p* = 0.001) and visceral fat (*p* = 0.003) compared to the DT group.
**Park et al. [53]**	Obese women (n = 11, 44.45 ± 0.8 yr, BMI > 25 kg/m^2^, %BF > 30%)	BFR	Frequency: 4 wks, 3 times/wk Intensity: 4 km/h, 5% grade Volume: 5 sets (each set 2 min) Type: Walk	160–230 mmHg	BW ↓ BMI ↓ FM ↓ TC↓	BW, BMI, FM, and TC decreased after exercise (*p* < 0.05).
**Chao Lan et al. [54]**	Healthy men (n = 50, 18–25 yr)	CON MICT HIIT LICT-BFR	Frequency: 8 wks, 3 times/wk Intensity: 57–63% HRmax Volume: 5 min, warm-up 15 min, 10 min resting phase Type: Walk	200–360 mmHg	%BF ↓ FM ↓ AVFA ↓	The LICT-BFR and MICT groups showed significant decreases in FM and AVFA compared to the CON group (*p* < 0.05). The exercise groups showed a significant decrease in %BF compared to the CON group (*p* < 0.05).
**Du-Hwan Oh et al. [55]**	Obese women (n = 11, 44.45 ± 0.8 yr, BMI > 25 kg/m^2^, %BF > 30%)	BFR	Frequency: 4 wks, 3 times/wk Intensity: 4 km/h, 5% grade Volume: 5 sets (each set 2 min) Type: Walk	160–230 mmHg	BW ↓ BMI ↓ %BF ↔	BW and BMI decreased significantly after exercise (*p* = 0.022, *p* = 0.015, respectively). %BF tended to decrease, but this change was not statistically significant (*p* = 0.07).

↓ indicates significantly decreased; ↔ indicates no significant difference; AVFA, abdominal visceral fat areas; BFR, blood flow restriction; BMI, body mass index; BW, body weight; %BF, body fat percentage; CON, control; DT, dual task; FM, fat mass; HIIT, high-intensity interval training; HRmax, heart rate max; HRR, heart rate reserve; LIT, low-intensity training; TC, thigh circumference; VO_2_ max, maximal oxygen consumption; WC, waist circumference.

## 5. Effect of Blood Flow Restriction Aerobic Exercise on Muscle Mass and Strength

Sarcopenia has recently been classified as a disease that affects not only physical performance, but also mortality in older adults. Furthermore, muscle strength has been identified as a risk factor for CVD and mortality, while decreased muscle mass is associated with various metabolic problems [56,57]. Consequently, the risk of decreased muscle mass and strength, along with increased body fat mass, has been increasingly recognized in recent years [57,58]. Resistance exercise is one of the most effective interventions for improving muscle hypertrophy and strength. However, resistance exercise requires a high mechanical load, which may limit its applicability to beginners or individuals with low fitness levels. Additionally, it may be challenging to induce muscle hypertrophy with classical aerobic exercise [59,60]. 

To date, many studies have demonstrated that BFRRE is capable of inducing muscle hypertrophy and strength gains. In this context, several recent studies have shown that low-intensity BFRAE can lead to not only fat loss but also increases in muscle mass. In various subjects, including older adults and obese individuals, low-intensity, flow-restricted aerobic exercise has induced significant increases in muscle and lean mass, muscle cross-sectional area, and muscle strength. The overall results are summarized in Table 2.

Muscle hypertrophy can be induced by hormonal responses such as GH and IGF-1. IGF-1 promotes hypertrophy by activating the mTOR signaling pathway [61]. Restriction of blood flow to exercising muscles induces the accumulation of metabolic by-products, such as lactic acid and inorganic phosphate (Pi), in the blood, leading to metabolic stress [38,39]. Metabolic stress induced by BFRRE has been shown to increase GH and IGF-1, which are key hormones influencing muscle hypertrophy, and to activate the mTOR pathway, an important signaling mechanism in protein synthesis [7,25,37,62]. A recent study by Barjaste et al. examined hormones associated with muscle hypertrophy in acute BFRAE (40% of VO_2_ max). The study reported a significant increase in serum GH (*p* = 0.046) in the blood flow-restricted walking group compared to the non-restricted exercise group. Additionally, IGF-1 serum concentrations increased significantly only in the blood flow-restricted walking group (*p* = 0.001) [63]. Furthermore, a study of healthy young men reported an increase in serum GH (*p* < 0.01) immediately after a single bout of blood flow-restricted aerobic walking compared to a group without blood flow restriction [50]. 

Another potential mechanism could be cell swelling. The accumulation of metabolic by-products can increase blood flow in muscle tissue and induce cell swelling, which can suppress catabolism and promote anabolism by maintaining protein balance [29,64]. Furthermore, previous studies have reported significant decreases in whole-body protein breakdown and leucine oxidation rates, reflecting whole-body leucine flux and whole-body leucine oxidation [41,42]. Although research of transient increases in muscle volume and cell swelling is limited to acute BFRRE, it is important to note that both BFRRE and BFRAE share the commonality of restricting blood flow. Consequently, the possibility of transient cell swelling occurring in BFRAE cannot be definitively excluded. However, there is a lack of molecular insights into the cellular processes involved in the anabolism mediated by BFRAE, necessitating further investigation for confirmation. 

According to the size principle for neuromotor control, slow-twitch fibers are recruited first during exercise. As exercise intensity increases, fast-twitch fibers (type II muscle fibers) are progressively recruited. Recent studies have reported that blood flow restriction transiently reduces muscle oxygenation, allowing for the recruitment of fast-twitch muscle fibers even at very low-intensity due to high metabolite accumulation [65,66,67]. In this respect, blood flow restriction induces a hypoxic environment within the muscle, which may lead to a reliance on anaerobic metabolism. Both reduced oxygen concentration and metabolite accumulation can accelerate neuronal fatigue, causing the human nervous system to mistakenly perceive the exercise as more intense than it actually is. Therefore, more muscle fibers are stimulated, leading to rapid recruitment. It has been suggested that this phenomenon may mechanically increase myofibrillar recruitment via afferent nerve stimulation, and also that increased neuro-fatigue may inhibit alpha motor neurons, thereby increasing myofibrillar recruitment as a compensatory response to maintain muscle strength and protect against conduction disturbances during exercise [68,69]. Specifically, this possibility is supported by literature showing higher motor unit recruitment and electromyographic (EMG) activation of fast-twitch muscle fibers during low-intensity BFRRE compared to the same exercise protocol without blood flow restriction [65,70,71,72]. However, concrete evidence that BFRAE relies on anaerobic metabolism during exercise to promote muscle hypertrophy through fast-twitch fiber mobilization is still lacking.

In summary, further research is needed to investigate the specific mechanisms that drive the hypertrophic effects of BFRAE. While resistance exercise has proven an effective intervention for older adults with sarcopenia, it can increase the risk of injury without professional supervision. Therefore, BFRAE may serve as an effective alternative exercise intervention for older adults and individuals with limitations that prevent them from engaging in resistance forms of exercise.

**Table 2 ijms-25-09274-t002:** Effects of aerobic exercise with blood flow restriction on muscle mass and strength.

Author	Subjects	Group	Intervention	Cuff Pressure	Outcomes	*p*-Value
**Abe et al. [73]**	Old adults (n = 19, 60–78 yr)	CON BFR	Frequency: 6 wks, 5 times/wk Intensity: 67 m/min Volume: 20 min Type: Walk	160–200 mmHg	Muscle strength ↑ Mid-thigh CSA ↑ Lower leg CSA ↑ Thigh muscle mass ↑	The BFR group showed significant increases in isometric (11%) and isokinetic (7–16%) knee extension and flexion torque, muscle CSA (5.8% for the thigh and 5.1% for the lower leg), as well as muscle mass (6.0% and 10.7% for total and thigh, respectively) (*p* < 0.05), but there was no significant difference in the CON group.
**Amir Kargaran et al. [52]**	Old women (n = 24, 62.9 ± 3.1 yr)	CON DT DT + BFR	Frequency: 8 wks, 3 times/wk Intensity: 45% HRR Volume: 20 min Type: Walk	150–200 mmHg	Muscle strength ↑ Muscle quality ↑	The DT + BFR group showed significant increases in muscle strength (*p* < 0.001) and muscle quality (*p* < 0.001) compared to both the DT and CON groups.
**Hayao Ozaki et al. [74]**	Old adults (n = 23, 57–76 yr)	MIT MIT + BFR	Frequency: 10 wks, 4 times/wk Intensity: 45% HRR Volume: 20 min Type: Walk	140–200 mmHg	Muscle CSA ↑ Muscle strength ↑	Muscle strength (∼15%) and Muscle CSA (3%) increased in the MIT + BFR group, with no significant difference observed in the MIT group.
**Abe et al. [75]**	Healthy men (n = 19, 20–26 yr)	LIT LIT + BFR	Frequency: 8 wks, 3 times/wk Intensity: 40% VO_2_ max Volume: 15 min Type: Cycle	160–210 mmHg	Muscle CSA ↑ Muscle volume ↑ Isometric muscle strength ↔	Muscle CSA and muscle volume increased by 3.4-5.1% (*p* < 0.01) in the BFR group, and isometric strength tended to increase by 7.7% (*p* = 0.10). In contrast, the CON group showed no significant difference in muscle size (~0.6%) and strength (~1.4%).
**Abe et al. [50]**	Healthy men (n = 18, 21.2 ± 2.7 yr)	CON BFR	Frequency: 3 wks, 2 times/day Intensity: 50 m/min Volume: 20 min Type: Walk	160–230 mmHg	Muscle CSA ↑ Muscle volume ↑ Muscle strength ↑	The BFR group showed increases in muscle CSA, muscle volume by 4–7%, and isometric strength by 8–10%. In contrast, the CON group showed no significant difference in muscle size or isometric strength.
**Mikako Sakamaki et al. [76]**	Healthy men (n = 31, 21.2 ± 1.9 yr)	CON BFR	Frequency: 3 wks, 6 times/wk, 2 times/day Intensity: 50 m/min Volume: 5 sets (each set 2 min, 1 min rest) Type: Walk	160–230 mmHg	Muscle volume ↑	The BFR group showed significant increases in upper leg muscle volume (3.8%, *p* < 0.05) and lower leg muscle volume (3.2%, *p* < 0.05). In contrast, there was no significant difference in muscle volume in the CON group.
**Kim et al. [9]**	Healthy men (n = 31, 22.4 ± 3.0 yr)	CON MIT LIT + BFR	Frequency: 6 wks, 3 times/wk Intensity: MIT: 60–70% HRR LIT + BFR: 30% HRR Volume: 20 min Type: Cycle	160–180 mmHg	Muscle strength ↑ Leg lean mass ↑	The MIT (7.15%) and LIT-BFR (8.90%) groups showed significant increases in muscle strength (*p* = 0.024 and *p* = 0.01, respectively). In contrast, there was no difference in muscle strength in the CON group. The LIT-BFR group increased the leg lean mass by 1.15% (*p* = 0.024), whereas there were no differences in leg lean mass in the MIT (0.44%) and CON (0.89%) groups.
**Oliveira et al. [77]**	Adults (n = 37, 23.8 ± 4.0 yr)	LIT LIT + BFR HIT HIT + BFR	Frequency: 4 wks, 3 times/wk Intensity: LIT + BFR: ~30% P_MAX_ HIT + BFR: ~66% P_MAX_ HIT: ~102% P_MAX_ Volume: 20 min Type: Cycle	140–200 mmHg	Muscle strength ↑	The LIT + BFR group increased isometric strength by 11.4 ± 7.3%, (*p* = < 0.001). However, there were no differences in isometric strength in other groups: HIT (−0.7 ± 9.9%, *p* = 0.88), HIT + BFR (−3.5 ± 6.8%, *p* = 0.32), and LIT (−2.6 ± 6.7%, *p* = 0.82).
**Beak et al. [8]**	Healthy men (n = 30, 30.21 ± 3.0 yr)	LIT LIT + BFR	Frequency: 8 wks, 3 times/wk Intensity: 40% VO_2_ max Volume: 15 min Type: Walk	160–240 mmHg	Muscle mass ↑ Muscle volume ↑	The LIT + BFR group showed increases in muscle mass (*p* <0.001) and right thigh circumference (*p* = 0.042), whereas no difference were observed in the LIT group.
**Park et al. [53]**	Obese women (n = 11, 44.45 ± 0.8 yr, BMI > 25 kg/m^2^, %BF > 30%)	BFR	Frequency: 4 wks, 3 times/wk Intensity: 4 km/h, 5% grade Volume: 5 sets (each set 2 min) Type: Walk	160–230 mmHg	Muscle strength ↑ Muscle endurance ↑	Muscle strength increased at 60°/s for right and left side extension, left side flexion, and at 180°/s for left side extension after training (*p* < 0.05).
**Chao Lan et al. [54]**	Health men (n = 50, 18–25 yr)	CON MICT HIIT LICT-BFR	Frequency: 8 wks, 3 times/wk Intensity: 57–63% Hrmax Volume: 5 min, warm-up 15 min, 10 min resting phase Type: Walk	200–360 mmHg	Muscle mass ↑	The LICT-BFR and MICT groups showed a significant increase in muscle mass (*p* < 0.05), while the HIIT group exhibited only a slight increased (*p* = 0.247) and the CON group showed a decrease in (*p* = 0.11).

↑ indicates significantly increased; ↔ indicates no significant difference; BFR, blood flow restriction; CON, control; CSA, cross sectional area; DT, dual task; HIIT, high-intensity interval training; HRmax, heart rate max; HRR, heart rate reserve; LIT, low-intensity training; MIT, moderate-intensity training; P_MAX_, maximal power output; VO_2_ max, maximal oxygen consumption.

## 6. Effects of Blood Flow Restriction Aerobic Exercise on Lipid Profiles and Glycemic Metabolism

Cholesterol, triglycerides, insulin, and blood glucose are important health indicators and representative risk factors for CVD [78,79,80,81]. Recent studies have reported improvements in blood glucose, insulin, cholesterol, and triglyceride levels with chronic BFRAE. A comprehensive summary of these results is presented in Table 3.

Blood glucose and insulin levels are regulated by adenosine monophosphate-activated protein kinase (AMPK) [82]. AMPK is a cellular energy sensor activated by an increase in the AMP/ATP ratio. It is activated in both rodents and humans during exercise and is known to increase in response to stressors such as hypoxia and ischemia [83,84,85,86,87]. AMPK plays a beneficial role in the regulation of glucose, lipid, and protein metabolism. It is involved in the activation of glucose transporter 4 (GLUT4) in muscle and the survival of β cells in the pancreas, as well as in improving insulin resistance [88,89].

Recent studies have shown that moderate-intensity BFRAE in trained men significantly increased muscle AMPK and PGC-1α mRNA expression compared to the non-restricted group [90]. Moreover, when compared to the restricted and non-restricted leg groups, the restricted leg group showed increased expression of AMPK and GLUT4 proteins in the muscle [91]. Although the majority of research on chronic BFRAE in both obese and healthy individuals has demonstrated improvements in blood glucose and glucose uptake, the study by Oh et al. reported different results [55]. These inconsistent outcomes may be attributed to factors such as the subjects’ gender, age, fitness level, and exercise regimen. Additional studies will be necessary to establish conclusive evidence. In particular, gender differences in glycemic metabolism have been reported with glycemic metabolism being highly responsive to physiological and nutritional states and fitness states [92]. Compared to men, women typically have lower skeletal muscle mass, higher fat mass, more circulating free fatty acids, and higher intracellular lipid content, which can affect glycemic metabolism [93]. In addition, the absence of estrogen and excess testosterone may lead to the development of type 2 diabetes mellitus in women [94]. Although available evidence is limited, BFRAE may potentially induce positive changes in blood glucose and insulin levels, which are risk factors for CVD, by activating AMPK-related signaling pathways in the muscles. These results suggest that BFRAE could be an effective intervention for improving insulin sensitivity and glucose metabolism through increased expression of AMPK and GLUT4 in muscle tissue.

BFRAE may positively affect blood glucose and lipid levels; however, conflicting findings have been reported regarding its effects on low-density lipoprotein cholesterol, high-density lipoprotein cholesterol, and total cholesterol. Furthermore, the mechanisms behind these effects are still unclear and insufficiently explored. More research is necessary to elucidate how BFRAE impacts blood cholesterol levels across diverse populations and with larger sample sizes. Additionally, understating the signaling pathway mechanism through molecular studies is crucial. 

**Table 3 ijms-25-09274-t003:** Effects of blood flow restriction aerobic exercise on lipid profiles, glycemic metabolism.

Author	Subjects	Group	Intervention	Cuff Pressure	Outcomes	*p*-Value
**Shuoqi Li et al. [48]**	Obese adults (n = 72, < 25 yr, %BF > 30%)	CON HIIT HIIT + BFR (during interval) HIIT + BFR (during exercise)	Frequency: 12 wks, 2 times/wk Intensity: 85% VO_2_ max Volume: 4 sets (each set 3 min, 3 min rest) Type: HIIT	40% limb occlusive pressure (LOP)	GLU ↓ Insulin ↓	The HIIT + BFR (during interval) and HIIT + BFR (during exercise) groups showed decreased blood GLU and insulin levels compared to the HIIT group (*p* < 0.05).
**Youn Chen et al. [51]**	Obese men (n = 40, 18–22 yr, %BF > 25% or BMI > 28 kg/m^2^)	LIT LIT + BFR	Frequency: 12 wks, 2 times/wk Intensity: 40% VO_2_ max Volume: 3 sets (each set 15 min, 1 min rest) Type: Cycle	200 mmHg	GLU ↓ TC ↓ TG ↔ LDL-C ↓ HDL-C ↑	The LIT + BFR group showed improved GLU, TC, HDL-C and LDL-C compared to the LIT group (*p* < 0.05).
**Omid Razi et al. [95]**	Obese men (n = 18, 37–55 yr, BMI = 27–28 kg/m^2^)	LIT LIT + BFR	Frequency: 8 wks, 3 times/wk Intensity: 3 km/h Volume: 5 sets Type: Walk	140–200 mmHg	TG ↓ LDL-C ↔ HDL-C ↔	The LIT + BFR and LIT groups showed a decrease in TG (*p* = 0.017). However, there were no differences in LDL-C and HDL-C between the LIT + BFR and LIT groups.
**Danny Christiansen et al. [91]**	Healthy men (n = 13, 25 ± yr)	CON leg BFR leg	Frequency: 6 wks, 3 times/wk Intensity: 60–80% Wmax Volume: 9 sets (each set 2 min, 2 min rest) Type: Cycle	180 mmHg	Glucose uptake ↑	Thigh net glucose uptake was higher in the BFR leg compared to the CON leg (*p* < 0.01)
**Oh et al. [55]**	Obese women (n = 11, 44.45 ± 0.8 yr, BMI > 25 kg/m^2^, %BF > 30%)	BFR	Frequency: 4 wks, 3 times/wk Intensity: 4 km/h, 5% grade Volume: 5 sets (each set 2 min, 1 min rest) Type: Walk	160–230 mmHg	GLU ↔ Insulin ↓ HOMA-IR ↓	Insulin and HOMA-IR decreased after exercise (*p* = 0.03, *p* = 0.04, respectively).

↓ indicates significantly decreased; ↑ indicates significantly increased; ↔ indicates no significant difference; BFR, blood flow restriction; BMI, body mass index; CON, control; GLU, glucose; HDL-C, high-density lipoprotein cholesterol; HIIT, high-intensity interval training; HOMA-IR, homeostatic model assessment for insulin resistance; LDL-C, low-density lipoprotein cholesterol; LIT, low-intensity training; TC, total cholesterol; TG, triglycerides; VO_2_ max, maximal oxygen consumption; Wmax, maximal work capacity.

## 7. Effects of Blood Flow Restriction Aerobic Exercise on Cardiovascular Function

The prevalence of CVD, a leading cause of mortality worldwide, has been consistently increasing [96]. Various methods for assessing cardiovascular function such as the pulse wave velocity (PWV), arterial compliance, heart rate variability (HRV), flow-mediated dilation, and other indicators are available. These factors are closely associated with the risk of CVD [97,98,99]. BFRAE has been shown to improve measures such as femoral-tibial PWV, venous compliance, systolic blood pressure (SBP), and HRV, as presented in Table 4.

In this review, chronic BFRAE appeared to have a positive impact on cardiovascular function; however, studies are limited. Most studies have focused on an acute session, and the acute effects of aerobic exercise with restricted blood flow have been shown to increase SBP, diastolic blood pressure, heart rate, and cardiac output (CO) [100,101,102].

Improving cardiovascular function through BFRAE may be related to the muscle metaboreflex. The muscle metaboreflex is a reflex response activated when the accumulation of metabolic by-products and depletion of oxygen in the muscle exceed the metabolic demands of the muscle [103,104]. Increased sympathetic nerve activity and decreased parasympathetic nerve activity in the cardiovascular system cause vasoconstriction [105]. Consequently, the temporary increase in heart rate, blood pressure, and CO observed during blood flow restriction may result from decreased oxygen supply and increased metabolite accumulation within the muscle. Previous research indicates five weeks of resistance training with blood flow restriction in the leg resulted in a significant decrease in the average arterial pressure, heart rate, and metabolic reflection within the muscle compared to the non-blood flow restriction leg [106]. These results suggest that long-term BFRAE can reduce the adaptive response of muscle to accumulated metabolites and sensitivity to signaling molecules. Therefore, the decrease in SBP after chronic BFRAE may be due to changes in metabolic reflection within the muscle.

Aerobic exercise is well known to have a positive effect on cardiovascular function. When blood flow increases during exercise, shear stress (i.e., the frictional force of blood on the arterial wall) occurs in the vascular endothelium, activating endothelial nitric oxide synthase (eNOS) and increasing NO production, which can improve vascular function [107,108]. One potential mechanism for the improvement in vascular function by BFRE is shear stress. The temporary increase in blood flow that occurs when blood flow restrictions are released can lead to an increase in blood flow, causing increased reactive hyperemic blood flow and increased venous return. Specifically, a previous study reported an increased reactive hyperemia index and transcutaneous oxygen pressure after four weeks of BFRE, indicating improved peripheral blood circulation compared to non-BFR in healthy older adults [109]. These results suggest that shear stress and NO production positively affect vascular function. 

Hypoxia is another potential mechanism. BFRE reduces blood flow to the exercising muscle, which induces a hypoxic environment. Hypoxia can increase the expression of hypoxia-inducible factor 1 subunit alpha (HIF-1α), which stimulates the expression of vascular endothelial growth factor (VEGF) and eNOS [110]. Barjaste et al. reported an increase in HIF-1α and VEGF protein expression after acute exercise at 40% of VO_2_ max compared to a group without blood flow restriction [63]. However, to date, results regarding the effect of chronic BFRAE on cardiovascular function remain inconsistent. This discrepancy may be attributed to variations in subjects’ fitness level, age, and training protocol. While the increased expression of VEGF, HIF-1α, and eNOS mRNA by vasculogenic progenitor cells following BFRRE has been demonstrated, research confirming the vasodilatory and angiogenic factors in response to BFRAE is limited [111,112]. Therefore, further studies are needed to investigate the effects of BFRAE on specific vascular function factors, such as NO, VEGF, and eNOS. 

**Table 4 ijms-25-09274-t004:** Effects of blood flow restriction aerobic exercise on cardiovascular function.

Author	Subjects	Group	Intervention	Cuff Pressure	Outcomes	*p*-Value
**Merat Karabulut et al. [113]**	Healthy men (n = 39, 18–50 yr)	CON HI LI LI + BFR	Frequency: 6 wks, 3 times/wk Intensity: HI: 60–70% VO_2_R LIT: 30–40% VO_2_R LIT + BFR: 30–40% VO_2_R Volume: 30 min Type: Walk	120 individuals’ thigh circumference: <45–50 cm = 120 mmHg 51–55 cm = 150 mmHg 56–59 cm = 180 mmHg ≥65 cm = 210 mmHg	Femoral–tibial PWV ↓ SBP ↔ DBP ↔	The LI + BFR group decreased the femoral–tibial PWV (*p* < 0.05) from baseline to post-training.
**Ozaki et al. [74]**	Sedentary adults (n = 23, 57–76 yr)	MIT MIT + BFR	Frequency: 10 wks, 4 times/wk Intensity: 45% HRR Volume: 20 min Type: Walk	140–200 mmHg	Carotid arterial compliance ↔	Carotid arterial compliance improved in both the MIT + BFR (50%) and MIT (59%) groups.
**Adalberto Ferreira et al. [114]**	Men (n = 21, 52.4 ± 3.7 yr)	CON BFR	Frequency: 6 wks, 3 times/wk Intensity: 6 km/h, 5% Volume: 5 sets (each set 3 min, 1 min rest) Type: Walk	80–100 mmHg	SBP ↓ DBP ↔ SDNN ↑ RMSSD ↑	Only the BFR group showed improvements in SDNN (*p* = 0.002), RMSSD (*p* = 0.01), and SBP (*p* = 0.006).
**Haruko Iida et al. [115]**	Women (n = 16, 59–78 yr)	CON BFR	Frequency: 6 wks, 5 times/wk Intensity: 67 min/m Volume: 20 min Type: Walk	140–200 mmHg	Venous compliance ↑	The BFR group showed improved venous compliance (*p* < 0.05), while there was no significant difference in the CON group.
**Beak et al. [8]**	Healthy men (n = 30, 30.21 ± 3.0 yr)	LIT LIT + BFR	Frequency: 8 wks, 3 times/wk Intensity: 40% VO_2_ max Volume: 15 min Type: Walk	160–240 mmHg	FMD ↔ baPWV ↔ SBP ↔ DBP ↔	There were no differences in the time × group interaction effects on FMD, baPWV, SBP, and DBP.

↓ indicates significantly decreased; ↑ indicates significantly decreased; ↔ indicates no significant difference; BFR, blood flow restriction; CON, control; DBP, diastolic blood pressure; HRR, heart rate reserve; LIT, low intensity training; MIT, moderate intensity training; PWV, pulse wave velocity; baPWV, brachial ankle pulse wave velocity; RMSSD, root mean square of the successive differences; FMD, flow-mediated dilation; SBP, systolic blood pressure; SDNN, standard deviation of NN intervals; VO_2_R, VO_2_ reserve.

## 8. Safety and Considerations for Blood Flow Restriction Aerobic Exercise

BFRE is a method of performing exercise by wearing a pressure cuff on the limbs. While it offers various physiological benefits, it may also cause side effects. Early studies reported potential risks when BFRE was applied to patients with conditions such as hypertension, heart failure, and peripheral arterial disease, including side effects such as rhabdomyolysis, pulmonary embolism, and venous thrombosis [116,117,118]. Therefore, concerns about the safety of applying BFRE to individuals with chronic metabolic disease and CVD are valid.

Despite these concerns, BFRE has been reported to be safe, particularly in older adults and patients with chronic conditions. In a recent survey of 136 BFRE experts, tingling (72.1%) and delayed-onset muscle soreness (55.8%) were the most commonly observed side effects, while cases of rhabdomyolysis, fainting, and subcutaneous hemorrhage were identified at a low frequency (1.9%, 3.8%, and 4.8%, respectively) [119]. Additionally, a study involving young adults (<20 years) and older adults (>80 years) reported very low rates of side effects such as deep vein thrombosis (0.055%), pulmonary embolism (0.008%), and rhabdomyolysis (0.008%) [120]. Notably, the study included patients with cerebrovascular diseases, obesity, CVD, diabetes, hypertension, and respiratory diseases. Furthermore, BFRE induced a reduction in SBP post-exercise at 60 minutes in adult women with hypertension, and a recent systematic review reported that BFRE can be safely performed without side effects in patients with hypertension as well as various types of heart disease and heart failure [121,122,123].

Furthermore, despite appropriate pressure settings, negative side effects such as discomfort, pain, numbness, and subcutaneous bleeding can occur, particularly in novice users who lack experience with BFRE [119]. These side effects are usually temporary and may disappear as the body adapts to the pressure cuff, with numbness typically returning to a normal state after exercise [124]. Therefore, it is essential to closely monitor participants during exercise in real time and to ensure that any adverse events are promptly managed with adequate rest and professional medical intervention.

However, to prevent side effects and ensure safety, it is necessary to consider various factors such as the degree of blood flow restriction (cuff pressure, width, and restriction time) for each participant, training variables (intensity, amount, and frequency), and medical history related to cardiovascular factors (CVD, anticoagulant medication, thrombosis, uncontrolled hypertension, etc.). It is crucial to use appropriate pressure, as excessive pressure during prolonged restriction can lead to thrombosis and venous damage. Cuff pressure should be individualized rather than applied on an absolute basis. Studies suggest that 40–90% of an individual’s resting arterial occlusive pressure (AOP) is an appropriate range [125,126]. Counts et al. demonstrated that either 40% or 90% arterial occlusion produced similar increases in muscle size, strength, and endurance after eight weeks of training [126]. Thus, using 40% of the arterial occlusion may help to minimize the shear rate produced by BFRE. Additionally, the width of the pressure cuff can influence the degree of arterial occlusion. Narrow cuffs produced lower ratings of perceived discomfort than wider cuffs when exercising at equal pressures; despite the same AOP, a greater decrease in blood flow was reported as the cuff width (5, 10, and 15 cm) increased [127,128]. Hence, it is important to set appropriate personalized restriction pressures and avoid full arterial occlusion due to unnecessarily high pressures.

However, the optimal conditions for chronic BFRAE in terms of frequency, intensity, and cuff pressure remain unclear. Therefore, it is necessary to establish accurate guidelines to ensure safety in the aging population and in those with CVD and hypertension. Efforts should be made to provide guidelines for the different variables that should be considered in blood flow restriction methods and to confirm their safety in future studies. Additionally, the time to restrict blood flow during resistance exercise is generally 5–10 minutes, while aerobic exercise is relatively longer, up to 20 min, so it is necessary to pay more attention. 

To summarize, individualized cuff pressure settings and pressure adjustments based on training variables, cuff width, and potential risk factors may be necessary when implementing BFRE. Furthermore, performing these exercises in an environment that allows real-time monitoring and immediate intervention may help to reduce the occurrence of adverse events and mitigate associated risk factors. Hence, future studies should provide guidelines for variables that should be considered in BFRAE, and efforts to verify its safety will be necessary.

## 9. Conclusions and Future Directions

This review summarizes the current evidence regarding the effects of BFRAE on body composition, muscle mass and strength, lipid and glycemic metabolism, and cardiovascular function. A summary of the potential effects of chronic BFRAE and the mechanisms involved is presented in Figure 1. BFRAE can lead to a reduction in adipose tissue, increase in muscle mass and strength, and positive effects on lipid and glucose metabolism and cardiovascular function. However, the studies presented in this review vary not only in the training variables (intensity, frequency, volume, duration, and cuff pressure) but also in the study populations (obese people, older adults, healthy adults, etc.). These differences in study design may be insufficient to demonstrate definitive conclusions regarding the research objectives of this review. Furthermore, the lack of time points measured to determine the long-term effects of BFRAE may limit the ability to provide specific guidelines for conducting future research. Therefore, further research is needed to determine the definite long-term effects of BFRAE and to provide exercise guidelines that ensure safety.

BFRAE is an effective approach that appears to stimulate physiological adaptive responses, such as a reduction in body fat and improved muscle mass and strength. Furthermore, BFRAE can be utilized as a valuable approach to provide additional stimuli for various individuals regarding traditional aerobic exercise. One of its key advantages is that it offers an alternative exercise method for people who are unable to perform high-intensity exercise. However, the studies summarized in this review included not only older adults but also a significant number of obese and healthy young adults. This may be due to the tendency of researchers to recruit healthy younger populations to demonstrate the effectiveness of BFRAE, prioritizing the safety of the subjects. Therefore, further studies are needed to determine the effectiveness of BFRAE in older adults. 

Hence, considering individual characteristics, including age and medical history, specific guidelines must be established to ensure the effective and safe application of BFRAE, and additional research and validation are necessary to guarantee the safety and efficacy of BFRAE across various populations.

## Figures and Tables

**Figure 1 ijms-25-09274-f001:**
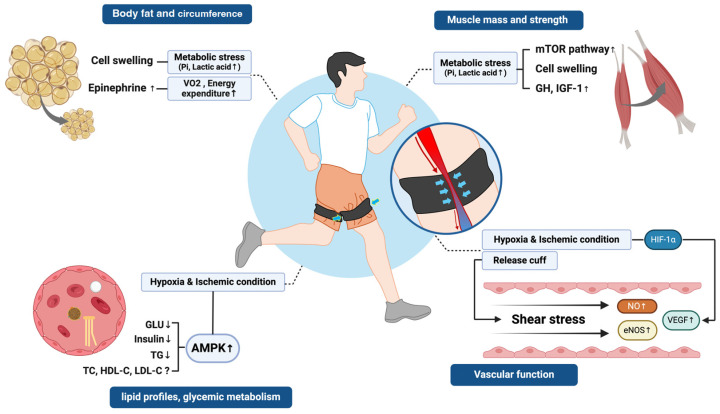
Physiological adaptive responses to chronic blood flow-restricted aerobic exercise. ↑, Indicates an increase in the corresponding protein and physiologic response; ↓ indicates significantly decreased; ?, insufficient evidence to provide a conclusion; eNOS, endothelial nitric oxide synthase; GH, growth hormone; GlU, glucose; HDL-C, high-density lipoprotein cholesterol; HIF-1α, hypoxia-inducible factor 1-alpha; IGF-1, insulin-like growth factor-1; LDL-C, low-density lipoprotein cholesterol; mTOR, mammalian target of rapamycin; NO, nitric oxide; Pi, inorganic phosphate; TC, total cholesterol; TG, triglyceride; VEGF, vascular endothelial growth factor; VO_2_, oxygen consumption.

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
