# Peer review of "The Effects of Blood Flow Restriction Aerobic Exercise on Body Composition, Muscle Strength, Blood Biomarkers, and Cardiovascular Function: A Narrative Review"

_ijms, 2024, doi:10.3390/ijms25179274_

Round 1

Reviewer 1 Report

Comments and Suggestions for Authors

This narrative review summarizes the effects of blood flow restriction (BFR) during aerobic exercise on various parameters. Given that most reviews on BFR focus on resistance exercise, this review provides new insights.

Overall:

BFR during exercise presents different physiological challenges compared to exercise without BFR. Therefore, it would be beneficial to include a section on the physiology and molecular signaling associated with BFR. This would provide a better physiological background before discussing each parameter (e.g., body composition, blood biomarkers). In the current version, the physiological aspects of BFR are scattered and need consolidation.

Throughout the text, when mentioning improvements (e.g., significantly increased), it is unclear whether these increases were relative to a control group or just observed improvements. Additionally, it is important to specify the population studied. Without this information, the reader does not get a clear picture, as any form of training with appropriate intensity can improve body composition, muscle mass, and strength in untrained individuals.

Throughout the text, please include the duration of the training and specify whether post-training measurements were taken acutely after the last session or a few days later. This will help determine whether the improvements are a direct response to exercise or an adaptation from training. Additionally, indicate if the studies mentioned in the tables incorporated progressive overloading.

For all tables, please include the percent change for the outcomes and clearly state whether the percent changes observed with BFR are compared to the pre-training measurements or other groups.

Section 2:
Citation #30 is mentioned in the second-to-last paragraph, but this study is not included in Table 1.

In the second-to-last paragraph of section 2, it is stated: "compared to high-intensity interval... body fat mass." Clarify whether the significant increases in GH and epinephrine were observed immediately after HIIT or at a basal state. When was the body fat mass measured? The current wording suggests simultaneous increases in GH, epinephrine, and fat mass reduction.

Physiologically, was the reduction in fat mass due to fat oxidation during exercise or an increase in basal fat oxidation?

In the last paragraph of section 2, the sentence does not accurately reflect the information presented in the previous paragraph (on HIIT rather than aerobic exercise). As previously mentioned, it is necessary to clarify what the changes are compared to.

Section 3:
Although the focus is on aerobic training with BFR, it is important to include information on BFR in resistance training (RT) concerning mass and strength changes. This will provide perspective on the effectiveness of BFRAE on mass and strength.

In the third paragraph, the sentence "Muscle hypertrophy... mTOR" is not entirely accurate. IGF-1 is the ligand that induces two major signaling cascades, one of which is mTOR.

Studies cited on BFR and GH involve resistance exercise, not aerobic exercise. This claim needs careful rephrasing, as the metabolic stress induced by BFR in RT and aerobic exercise might differ. The authors speculate that metabolic stress induced by BFR during AE could be comparable to RT, but this is unlikely.

The hypothesis that cell swelling suppresses catabolism and promotes anabolism is based on citations (42 and 43) related to muscle-damaging swelling and extreme dehydration. In the context of exercise-induced metabolic stress, it is questionable whether BFR stress can induce such significant swelling compared to high-volume, short-rest-period RT. This potential mechanism of action must be rephrased to accurately reflect current knowledge.

In the fourth paragraph: "As the intensity of exercise increases, fast... recruited." Does exercise with BFR increase exercise intensity or cause an earlier onset of fatigue due to lack of oxygen to type I fibers, necessitating type II fiber recruitment?

Please provide more information on how increased activation of anaerobic metabolism can increase mass and strength.

Regarding fat mass reduction, if less oxygen is supplied to type I fibers, there will be less fat oxidation. What mechanisms contribute to fat mass reduction, especially considering the hormonal response to exercise is acute and transient?

Last paragraph: If the goal is to prevent or delay the onset of type II fiber loss, would BFR with RT be more appropriate than AE? Without seeing the percent change in outcome variables and comparing these changes to BFR with RT, this statement is unsupported.

The legend for Table 2 is inaccurate and should not mention body composition.

Section 4: First paragraph: "Recent studies... with obesity." This sentence suggests that BFR with AE is not ideal for glucose regulation, insulin, etc.

Third paragraph: The change in AMPK and GLUT4 was observed in trained men. However, Table 3 shows that most studies using obese adults did not show improvements in GLUT4, insulin, etc. Please provide a potential mechanism for this discrepancy.

Table 3 is missing an author column.

Section 5: Fifth paragraph: "Additionally, ... cytokines... by BFRAE." This sentence appears abruptly. The mention of cytokines needs to be contextualized within the discussion.

Author Response

Title: The Effects of Blood Flow Restriction Aerobic Exercise on Body Composition, Muscle Strength, Blood Biomarker, and Cardiovascular Function: A Narrative Review

The authors express their gratitude to the reviewers for their valuable feedback. We have carefully examined and thoroughly addressed each comment to resolve all concerns raised. Please refer to the updated main text, with changes indicated in red. A detailed response is provided below. Thank you again for constructive reviews from the reviewers.

Response to Reviewer 1

Comment 1.:

BFR during exercise presents different physiological challenges compared to exercise without BFR. Therefore, it would be beneficial to include a section on the physiology and molecular signaling associated with BFR. This would provide a better physiological background before discussing each parameter (e.g., body composition, blood biomarkers). In the current version, the physiological aspects of BFR are scattered and need consolidation.

Response 1:

Thank you sincerely for your constructive feedback. In response to comments from reviewers, the authors have incorporated new sections on physiological reactions to exercises that restrict blood flow and molecular signaling mechanisms at 2. Potential physiological and molecular mechanisms of blood flow restriction exercise.

Comment 2.:

Throughout the text, when mentioning improvements (e.g., significantly increased), it is unclear whether these increases were relative to a control group or just observed improvements. Additionally, it is important to specify the population studied. Without this information, the reader does not get a clear picture, as any form of training with appropriate intensity can improve body composition, muscle mass, and strength in untrained individuals.

Response 2: Thank you sincerely for your constructive feedback. The authors agree with the reviewers' comments and have revised the text overall.

Comment 3.:

Throughout the text, please include the duration of the training and specify whether post-training measurements were taken acutely after the last session or a few days later. This will help determine whether the improvements are a direct response to exercise or an adaptation from training. Additionally, indicate if the studies mentioned in the tables incorporated progressive overloading.

Response 3: We have revised the text based on the reviewer's comments. However, please understand that we cannot present studies that do not mention whether the measurement was performed acutely or several days after the last session. Additionally, no studies included progressive overload.

Comment 4.:

For all tables, please include the percent change for the outcomes and clearly state whether the percent changes observed with BFR are compared to the pre-training measurements or other groups.

Response 4: Thankyou sincerely for your constructive feedback. All tables included specific information reflecting the reviewers' comments, but papers that did not present the percentage change were expressed as p-values.

Comment 5.:

Section 2

Comment 5.1: Citation #30 is mentioned in the second-to-last paragraph, but this study is not included in Table 1.

Response 5.1: Citation #30 has been removed.

Comment 5.2: In the second-to-last paragraph of section 2, it is stated: "compared to high-intensity interval... body fat mass." Clarify whether the significant increases in GH and epinephrine were observed immediately after HIIT or at a basal state. When was the body fat mass measured? The current wording suggests simultaneous increases in GH, epinephrine, and fat mass reduction.

Response 5.2: The paragraph has been reorganized.

Comment 5.3: Physiologically, was the reduction in fat mass due to fat oxidation during exercise or an increase in basal fat oxidation?

Response 5.3: Based on the findings of the chronic BFRAE studies to date, it remains unclear whether the reduction in fat mass is influenced by fat metabolism during the exercise session or if it is the result of an enhanced capacity to oxidize basal fat. To further explore this, we recommend that future studies investigate the underlying mechanisms. This suggestion has been incorporated into the last paragraph of Section 2 in the main text.

Comment 5.4: In the last paragraph of section 2, the sentence does not accurately reflect the information presented in the previous paragraph (on HIIT rather than aerobic exercise). As previously mentioned, it is necessary to clarify what the changes are compared to.

Response 5.4: The last paragraph has been reorganized

Section 3:

Comment 6.1: Although the focus is on aerobic training with BFR, it is important to include information on BFR in resistance training (RT) concerning mass and strength changes. This will provide perspective on the effectiveness of BFRAE on mass and strength.

Response 6.1: We have included existing research on information on BFR in resistance training (RT) in this section.

Comment :6.2 In the third paragraph, the sentence "Muscle hypertrophy... mTOR" is not entirely accurate. IGF-1 is the ligand that induces two major signaling cascades, one of which is mTOR.

Response 6.2: The sentence has been reorganized.

Comment 6.3: Studies cited on BFR and GH involve resistance exercise, not aerobic exercise. This claim needs careful rephrasing, as the metabolic stress induced by BFR in RT and aerobic exercise might differ. The authors speculate that metabolic stress induced by BFR during AE could be comparable to RT, but this is unlikely.

Response 6.3: We added a study confirming the expression of growth hormone and IGF-1 in BFRAE.

Comment 6.4: The hypothesis that cell swelling suppresses catabolism and promotes anabolism is based on citations (42 and 43) related to muscle-damaging swelling and extreme dehydration. In the context of exercise-induced metabolic stress, it is questionable whether BFR stress can induce such significant swelling compared to high-volume, short-rest-period RT. This potential mechanism of action must be rephrased to accurately reflect current knowledge.

Response 6.4: An additional section (2. Potential physiological and molecular mechanism of blood flow restriction exercise) has been added to describe studies showing that it induces transient cell swelling.

Comment 6.5: In the fourth paragraph: "As the intensity of exercise increases, fast... recruited." Does exercise with BFR increase exercise intensity or cause an earlier onset of fatigue due to lack of oxygen to type I fibers, necessitating type II fiber recruitment?

Response 6.5: The content is described in detail in the 5th paragraph.

Comment 6.6: Please provide more information on how increased activation of anaerobic metabolism can increase mass and strength.

Response 6.6: The content is described in detail in the 5th paragraph.

Comment 6.7: Regarding fat mass reduction, if less oxygen is supplied to type I fibers, there will be less fat oxidation. What mechanisms contribute to fat mass reduction, especially considering the hormonal response to exercise is acute and transient?

Response 6.7: Thank you for this constructive question. Please understand that there is insufficient evidence to provide a definitive answer to the reviewer's question. BFRAE has been shown to induce transient increases in growth hormone and noradrenaline. Growth hormone is well known as a hormone that induces lipolysis, and noradrenaline is also known to play a key role in inducing lipolysis (PMID: 31780780/PMID: 30390616). In addition, aerobic exercise with blood flow restriction increases oxygen demand compared to aerobic exercise without blood flow restriction (PMID: 35357762, PMID: 34539445). This means that oxygen consumption is higher for exercise with blood flow restriction than for exercise without blood flow restriction. While these factors may influence fat loss, there is a paucity of research on whether blood flow restriction activates or inhibits fat metabolism concurrently with a hypoxic environment, and future studies should investigate the mechanisms of fat metabolism.

This content has been revised 3-5 paragraphs 'Section 3. Effects of blood flow restriction aerobic exercise on body composition' for reader’s understanding.

Comment 6.8: Last paragraph: If the goal is to prevent or delay the onset of type II fiber loss, would BFR with RT be more appropriate than AE? Without seeing the percent change in outcome variables and comparing these changes to BFR with RT, this statement is unsupported.

Response 6.8: We concur with the reviewer's assertion. This evaluation elucidated research findings indicating that blood flow restricted aerobic exercise leads to muscle hypertrophy. However, if the comparison is to be made here with blood flow restricted resistance exercise, the purpose of the review would be to compare the exercise form (resistance vs aerobic). The paragraph mentions that studies of blood flow restricted aerobic exercise have increased muscle mass and strength, suggesting that it may be an effective form of exercise for people who are limited in resistance exercise. Therefore, the sentence lacking direct evidence has been removed and the paragraph reorganized.

Comment 6.9: The legend for Table 2 is inaccurate and should not mention body composition.

Response 6.9: Revised the legend of the table to muscle mass and strength.

Section 4

Comment 7.1: Section 4: First paragraph: "Recent studies... with obesity." This sentence suggests that BFR with AE is not ideal for glucose regulation, insulin, etc.

Response 7.1: The sentence has been reorganized

Comment 7.2: Third paragraph: The change in AMPK and GLUT4 was observed in trained men. However, Table 3 shows that most studies using obese adults did not show improvements in GLUT4, insulin, etc. Please provide a potential mechanism for this discrepancy.

Response 7.2: Added content to the third paragraph.

Comment 7.3: Table 3 is missing an author column.

Response 7.3: Table 3 has been revised.

Section 5

Comment 8: Fifth paragraph: "Additionally, ... cytokines... by BFRAE." This sentence appears abruptly. The mention of cytokines needs to be contextualized within the discussion.

Response 8: The sentence has been removed to consider the context of the paragraph.

Reviewer 2 Report

Comments and Suggestions for Authors

The authors have reviewed the effects of BFRAE on somatic composition, muscle mass and strength, lipid profiles, glycaemic metabolism and cardiovascular function. They also review related safety considerations. Four tables of reviewed studies are included. The overall findings are summarised in a figure.

It is important that other groups be able to replicate these findings. At present, this is not the case, as other groups are likely to choose different publications. I strongly recommend that the authors completely re-write their submission, this time making it a systematic review. As part of this process, for each major parameter potentially related to BFRAE, the authors should list the databases and registers from which relevant studies were identified. The screening process for each such set of studies, including the eligibility criteria, should be detailed. A flow diagram for each should be included in the new submission. Each such diagram should give the numbers of databases and registers identified; the numbers removed before screening; the number of studies screened; the number excluded; the number sought for retrieval and the number not retrieved; the number assessed for eligibility; the number excluded, including the reasons for excluding each of these; and the numbers included in the review of that particular parameter.

Author Response

Title: The Effects of Blood Flow Restriction Aerobic Exercise on Body Composition, Muscle Strength, Blood Biomarker, and Cardiovascular Function: A Narrative Review

The authors express their gratitude to the reviewers for their valuable feedback. We have carefully examined and thoroughly addressed each comment to resolve all concerns raised. Please refer to the updated main text, with changes indicated in red. A detailed response is provided below. Thank you again for constructive reviews from the reviewers.

Response to Reviewer 2

Comment 1.:

It is important that other groups be able to replicate these findings. At present, this is not the case, as other groups are likely to choose different publications. I strongly recommend that the authors completely re-write their submission, this time making it a systematic review. As part of this process, for each major parameter potentially related to BFRAE, the authors should list the databases and registers from which relevant studies were identified. The screening process for each such set of studies, including the eligibility criteria, should be detailed. A flow diagram for each should be included in the new submission. Each such diagram should give the numbers of databases and registers identified; the numbers removed before screening; the number of studies screened; the number excluded; the number sought for retrieval and the number not retrieved; the number assessed for eligibility; the number excluded, including the reasons for excluding each of these; and the numbers included in the review of that particular parameter.

Response 1:

Thank you sincerely for your constructive feedback. The authors attempted to convert to a systematic review based on the reviewer's comments. Unfortunately, the authors concluded that switching to a systematic review was not an option to improve this paper's quality. The reasons for this are listed below.

  1. The authors utilized various search engines such as PubMed, Google Scholar, and Web of Science to search through the following keywords for the systematic review.

Topic

Key words

Table 1. Effects of aerobic exercise with blood flow restriction on body composition.

“blood flow restriction aerobic exercise or BFRAE” AND “treadmill or cycle or walking or running or aerobic exercise” AND “body fat or fat mass or body fat percent or body weight or visceral fat or circumference”

Table 2. Effects of aerobic exercise with blood flow restriction on muscle mass and strength.

“blood flow restriction aerobic exercise or BFRAE” AND “treadmill or cycle or walking or running or aerobic exercise” AND “muscle mass or muscle cross sectional area or muscle strength or muscle volume or muscle hypertrophy or lean mass”

Table 3. Effects of blood flow restriction aerobic exercise on lipid profiles, glycemic metabolism.

“blood flow restriction aerobic exercise or BFRAE” AND “treadmill or cycle or walking or running or aerobic exercise” AND “glucose or insulin or cholesterol or triglyceride”

Table 4. Effects of blood flow restriction aerobic exercise on cardiovascular function

“blood flow restriction aerobic exercise or BFRAE” AND “treadmill or cycle or walking or running or aerobic exercise” AND “pulse wave velocity or blood pressure or vascular function or vascular compliance or vascular stiffness or flow mediated dilation”

However, the number of studies was not sufficient to conduct a systematic review. In general, the number of articles to ensure the quality level of a systematic review or meta-analysis is more than 1000. However, the number of articles retrieved through the keywords was about 350 in total. In particular, the number of studies related to blood biomarkers and cardiovascular-related studies that met the criteria for systematic review was significantly lower. Furthermore, the number of articles is further reduced when studies without any mention of randomized controlled trials (RCTs) in the study design are excluded. In this respect, the representativeness of the study topic and objective may be reduced and the quality of the study as a systematic review cannot be guaranteed.

  1. In addition, when conducting a systematic review, the PICOS framework can be employed to enhance its precision. The framework specifies a particular population group, such as a specific age, gender, and medical condition, under the "Population" category. Nevertheless, this review encompasses studies on young and healthy individual’s adults, obese adults, and older adults. Moreover, the studies presented in this review vary in frequency, strength, and number. This makes it difficult to approach a systematic review and may not provide comprehensive and reliable evidence.

Taken together, it was determined that the pool of studies to conduct a systematic review is still insufficient and that converting the review to a systematic review would not guarantee the quality of the paper. Therefore, a narrative review would be a better way to provide a broad overview of the possible effects of blood flow restriction aerobic exercise on metabolic health markers, with the flexibility to incorporate a wide range of studies. The authors will continue to look for trends in future research on the topics covered in the current narrative review. Unfortunately, the limited number of studies on the topic led us to conduct a narrative review, but our final goal is to conduct a systematic and meta-analysis based on enough studies, which will help to support our current findings.

Reviewer 3 Report

Comments and Suggestions for Authors

In this study, Blood Flow Restriction Exercise (BFRE) is proposed as an exercise for elderly individuals and those unable to participate in high-intensity exercise. Among these, Blood Flow Restriction Aerobic Exercise (BFRAE) is presented for its effects on body composition, lipid profiles, glycemic metabolism, and cardiovascular function. Through a review of previous studies, the physiological effects of BFRAE have been shown to decrease fat mass, increase muscle mass, and enhance muscular strength, potentially benefiting lipid profiles, glycemic metabolism, and overall function. Additionally, the researcher suggests that BFRAE is suitable for individuals with low fitness levels, those prone to injury, the elderly, obese individuals, and those with metabolic disorders.

  • It is recommended to add the definitions and application methods of BFRE and BFRAE in the introduction for readers who have a low understanding of BFRE.
  • It is well known that using BFRE with resistance exercise is effective in increasing muscle mass and strength. Many previous studies have also shown that blood flow restriction aerobic exercise is effective on muscle mass and strength (Table 2). What do you think is the reason for similar effects despite the different main effects of resistance exercise and aerobic exercise?
  • Although BFRE is mainly applied to elderly individuals and those unable to participate in high-intensity exercise, many studies in Table 2 involve healthy subjects in their 20s. There must be a reason for conducting experiments on healthy subjects in those studies, and it is suggested to reflect the results in this paper as well.
  • Given that previous studies presented in the tables have different application periods, intensities, and repetitions of exercise, is it reasonable to interpret them uniformly?
  • It is necessary to observe symptoms such as pressure adjustment, discomfort, pain, and numbness as precautions during BFRE. Please add this part.
  • Since this study is a review of previous studies, it is suggested to add a meta-analysis.

Author Response

Title: The Effects of Blood Flow Restriction Aerobic Exercise on Body Composition, Muscle Strength, Blood Biomarker, and Cardiovascular Function: A Narrative Review

The authors express their gratitude to the reviewers for their valuable feedback. We have carefully examined and thoroughly addressed each comment to resolve all concerns raised. Please refer to the updated main text, with changes indicated in red. A detailed response is provided below. Thank you again for constructive reviews from the reviewers.

Response to Reviewer 3

Comment 1.:

It is recommended to add the definitions and application methods of BFRE and BFRAE in the introduction for readers who have a low understanding of BFRE.

Response 1:

Thank you sincerely for your constructive feedback. For readers, definitions and application methods for BFRE, BFRRE, and BFRAE have been added to the introduction.

Comment 2.:

It is well known that using BFRE with resistance exercise is effective in increasing muscle mass and strength. Many previous studies have also shown that blood flow restriction aerobic exercise is effective on muscle mass and strength (Table 2). What do you think is the reason for similar effects despite the different main effects of resistance exercise and aerobic exercise?

Response 2:

Thank you for the reviewer's question. I think a clear answer to the question can be revealed through additional mechanism studies in the future. The author's opinion is as follows.

The commonality between blood flow restriction aerobic exercise and blood flow restriction resistance exercise is that they restrict blood flow. This means that both forms of exercise are exposed to a hypoxic environment. The local hypoxic environment in the muscle leads to the accumulation of metabolites (lactate, hydrogen ions), which can activate signaling pathways that stimulate anabolic processes for protein synthesis in the muscle. Specifically, it can stimulate the mTOR signaling mechanism and hormones that induce muscle hypertrophy, such as IGF-1.

The hypoxic environment provides an environment in which oxygen availability in the muscle is reduced. This increases the utilization of fast-twitch fibers, which the body normally uses for high-intensity activities. Therefore, the enhanced recruitment of fast-twitch fibers can contribute to muscle hypertrophy and strength gains.

In summary, the mechanism shared by both types of exercise is thought to be the hypoxic environment, which may lead to increases in muscle mass and strength, and this could be confirmed through additional biochemical studies.

Comment 3.:

Although BFRE is mainly applied to elderly individuals and those unable to participate in high-intensity exercise, many studies in Table 2 involve healthy subjects in their 20s. There must be a reason for conducting experiments on healthy subjects in those studies, and it is suggested to reflect the results in this paper as well.

Response 3:

Thank you sincerely for your constructive feedback. As per the reviewer's opinion, we mentioned the reason why the study population comprised healthy young adults. We added it to the 'Conclusion' section of the paper, considering the flow of the manuscript.

Comment 4.:

Given that previous studies presented in the tables have different application periods, intensities, and repetitions of exercise, is it reasonable to interpret them uniformly?

Response 4:

Thank you sincerely for your constructive feedback. The authors agree with the reviewer's opinion. This review was written for the purpose of a narrative review, which is a review method that comprehensively reviews the research results on the effects of blood flow restriction aerobic exercise on metabolic health and describes them in an unstructured format, including the authors' personal interpretation and insights. In addition, the main purpose of this narrative review is to summarize the flow of research results and major findings on the effects of blood flow restriction aerobic exercise on metabolic health, and to suggest future research directions. Therefore, please understand that this review has limitations as a narrative review. Also, please understand that in the ‘Conclusion’ of this review, it was explained that additional studies are needed to clearly demonstrate the effects of blood flow restriction aerobic exercise.

A clear interpretation of the study purpose may be appropriate using a systematic and quantitative analysis of the results, such as meta-analysis. Perhaps this is why the reviewer suggested adding meta-analysis in the last comment. The response to this is described in Response 6.

In the future, the authors will conduct a meta-analysis through additional studies to provide basic data that can suggest optimal guidelines for the most effective intensity, frequency, repetitions, and pressure of blood flow restriction aerobic exercise.

Comment 5.:

It is necessary to observe symptoms such as pressure adjustment, discomfort, pain, and numbness as precautions during BFRE. Please add this part.

Response 5:

Thank you sincerely for your constructive feedback. In response to reviewer comments, we added caution regarding discomfort, pain, and numbness that may occur due to blood flow restriction to “Safety and considerations for blood flow restriction aerobic exercise.”

Comment 6.:

Since this study is a review of previous studies, it is suggested to add a meta-analysis.

Response 6:

To demonstrate a clear effect of blood flow-restricted aerobic exercise on metabolic health, a meta-analysis or systematic review can be performed to search for studies based on systematic and clear criteria and synthesize the results through statistical analysis to provide objective and reliable conclusions. The authors tried to convert to a meta-analysis based on the reviewers' comments. The authors screened through PubMed, Web of Science, and Google Scholar using the keywords presented in Table 1. Typically, the number of studies retrieved to perform a meta-analysis is more than 1000. However, the number of studies that confirmed changes in the four major metabolic health outcomes with blood flow-restricted aerobic exercise was significantly lower (approximately 350). This, in turn, may skew the results by reducing the reliability of the findings, such as reduced statistical power, heterogeneity issues, and poor generalizability. In particular, the number of studies related to blood biomarkers and cardiovascular function was significantly lower: for example, in the subdomains RMSSD, SBP, and venous compliance, which correspond to the main title, there should be at least 3-4 studies each, but the number of studies is 1-2. Therefore, please consider that the number of studies for the meta-analysis does not meet the number of studies to calculate an appropriate effect size.

Table 1

Topic

Key words

Table 1. Effects of aerobic exercise with blood flow restriction on body composition.

“blood flow restriction aerobic exercise or BFRAE” AND “treadmill or cycle or walking or running or aerobic exercise” AND “body fat or fat mass or body fat percent or body weight or visceral fat or circumference”

Table 2. Effects of aerobic exercise with blood flow restriction on muscle mass and strength.

“blood flow restriction aerobic exercise or BFRAE” AND “treadmill or cycle or walking or running or aerobic exercise” AND “muscle mass or muscle cross sectional area or muscle strength or muscle volume or muscle hypertrophy or lean mass”

Table 3. Effects of blood flow restriction aerobic exercise on lipid profiles, glycemic metabolism.

“blood flow restriction aerobic exercise or BFRAE” AND “treadmill or cycle or walking or running or aerobic exercise” AND “glucose or insulin or cholesterol or triglyceride”

Table 4. Effects of blood flow restriction aerobic exercise on cardiovascular function

“blood flow restriction aerobic exercise or BFRAE” AND “treadmill or cycle or walking or running or aerobic exercise” AND “pulse wave velocity or blood pressure or vascular function or vascular compliance or vascular stiffness or flow mediated dilation”

Round 2

Reviewer 2 Report

Comments and Suggestions for Authors

It is clearly too soon for a systematic review to be carried out in this area.

Author Response

None